# Assessment of the New Acceleromyograph TOF 3D Compared with the Established TOF Watch SX: Bland-Altman Analysis of the Precision and Limits of Agreement between Both Devices—A Randomized Clinical Comparison

**DOI:** 10.3390/jcm11154583

**Published:** 2022-08-05

**Authors:** Stefan Soltesz, Jan Thomas, Michael Anapolski, Guenter Karl Noé

**Affiliations:** 1Department of Anesthesia and Intensive Care Medicine, Rheinlandklinikum Dormagen, 41540 Dormagen, Germany; 2Department Ob/Gyn, Rheinlandklinikum Dormagen, University of Witten-Herdecke, 41540 Dormagen, Germany

**Keywords:** acceleromyography, neuromuscular block, neuromuscular monitoring

## Abstract

The new acceleromyograph TOF 3D was compared with the established TOF Watch SX in patients undergoing elective laparoscopic gynecological surgery. Neuromuscular transmission was assessed by simultaneous recording with both devices. Measurements were performed simultaneously at the left and the right M. adductor pollicis (Group A, 25 patients), or the M. corrugator supercilii (Group CS, 25 patients). The repeatability, time course, and limits of agreement (Bland-Altman) were compared. The primary endpoint was the 90% train-of-four recovery time (TOFR 0.9). Other endpoints included onset time of block, maximum T1 depression, time to 25% T1 recovery, the recovery time course of T1 response, and TOF ratio, respectively. In group CS, the repeatability coefficient of the TOF 3D was lower (4.66 (1.6)) than of the TOF Watch SX (6.02 (1.9); *p* = 0.026). In group A, the onset of the block was faster when measured by the TOF 3D (98.7 (30) s vs. 112.2 (36) s (mean (SD)); *p* = 0.032). In group A, time to recovery to a TOFR of 90% was measured earlier by the TOF 3D (bias −0.71 min, limits of agreement from −8.94 to +7.51 min). The TOF 3D provides adequate information with high precision and sensitivity. It is suitable even for measurement sites with small muscle contractions such as the M. corrugator supercilii.

## 1. Introduction

Intraoperative objective measurement of the neuromuscular block is an important tool to provide adequate surgical conditions during the procedure and to ensure the complete reversal of the neuromuscular block at the end of the operation. It is recommended as mandatory in all cases when using neuromuscular blocking agents [1,2]. Usually, a serial stimulation of four stimuli (train-of-four rate, or TOFR, respectively) is performed in a clinical setting. This technique offers the advantage that nearly all degrees of a neuromuscular block can be assessed objectively in the unconscious patient. However, in the clinical setting, reliable measurements might be challenging since the evoked responses of the muscles frequently are unstable and vary significantly. Additionally, they can be difficult to quantify, especially in the case of facial muscles, where the contractions are relatively small and therefore might be confounded with direct muscular stimulations [3].

Acceleromyography of the M. adductor pollicis is probably the technique used most frequently for this purpose in the clinical routine setting [1,4] because the amplitudes of the muscular response are relatively high, and direct muscle stimulation can be avoided easier than in the facial muscles. However, the positioning of the patient might restrict access to the arms making a measurement at the M. adductor pollicis impossible.

The TOF 3D is a new acceleromyograph designed to assess the degree of a neuromuscular block during anesthesia. According to the manufacturer, this new device is able to record the movements of the sensor in three dimensions. Thus, even responses with very small amplitudes could be detected.

The study compared the new three-dimensional technique of the TOF 3D with the established TOF Watch^®^ SX (Essex Pharma GmbH, Munich, Germany) with regard to repeatability (=precision), the time course of the neuromuscular block, and limits of agreement. We chose two different sites for nerve stimulation: In the first part of the investigation, both devices were compared during stimulation of the M. adductor pollicis, because this stimulation site probably is most frequently used in the clinical routine setting.

In the second part, we assessed the performance of the acceleromyographs by measuring the response of the M. corrugator supercilii: this site of stimulation is more challenging because of the difficulties already mentioned above. Possibly, quality differences between the devices would be more pronounced at this stimulation site.

Measurements were performed simultaneously with both devices at the M. adductor pollicis of both arms, or at the M. corrugator supercilii on both sides.

The primary outcome parameter was the difference in the return to a train-of-four Rate of 0.9 (TOFR 0.9) between both devices. Secondary outcome parameters were differences with regard to the time course of the neuromuscular block, repeatability, bias, limits of agreement (Bland-Altman analysis) [5,6], and the Surgical Rating Scale (SRS) [7,8].

## 2. Materials and Methods

### 2.1. Ethics

After obtaining local ethics committee approval (Ethikkommission der Universität Witten/Herdecke, Alfred-Herrhausen-Str. 50, 58448 Witten, Germany; 7 August 2021; chairman Prof. Dr. Peter Gaidzik) and following patients’ written informed consent, we undertook this prospective, unblinded, single center randomized study (ClinicalTrial.gov Identifier: NCT05005676).

### 2.2. Patient Selection

We recruited female patients aged 18 to 65 years, American Society of Anesthesiologists physical status I–III, 50–90 kg body weight, undergoing elective laparoscopic gynecological surgery (n = 50).

Exclusion criteria included expected difficult tracheal intubation (i.e., history of prior difficult intubation, reduced opening of the mouth (<2 cm), Mallampati Score 4), increased risk of pulmonary aspiration (i.e., gastroesophageal reflux, full stomach, intestinal obstruction), known allergies to anesthetic drugs, pregnancy, neuromuscular disorders, intake of drugs affecting neuromuscular block (e.g., furosemide, magnesium or cephalosporins), and hepatic (serum bilirubin > 26 µmol L^−1^) or renal (serum creatinine > 90 µmol L^−1^) insufficiency.

### 2.3. Induction and Maintenance of Anesthesia

Patients were pre-medicated with midazolam 7.5 mg per os and pre-oxygenated with 100% O_2_ via a tightly fitted facemask. Anesthesia was induced with an intravenous infusion of remifentanil 0.2 µg kg^−1^ min^−1^, single bolus doses of sufentanil 0.2–0.3 µg kg^−1^, and propofol 2–3 mg kg^−1^. Initially, a laryngeal mask (Ambu^®^ AuraOnce™ size 4, Ambu Inc., Glen Burnie, MD, USA) was inserted, and anesthesia was maintained by infusion of remifentanil 0.15–0.25 µg kg^−1^ min^−1^ and propofol 3–5 mg kg^−1^ h^−1^. The laryngeal mask was eventually replaced by an endotracheal tube (Rueschelit^®^ 7.5 mm I.D., TeleflexMedical, Athlone, Ireland) after the establishment of the neuromuscular block (below). Patients’ lungs were mechanically ventilated with a tidal volume of 6–8 mL kg^−1^. The respiratory rate was adjusted until normocapnia (end-tidal CO_2_ concentration 36–40 mm Hg) was achieved. Heart rate and systemic arterial blood pressure were maintained within ±20% of baseline values during the study. Nasopharyngeal and skin surface temperatures (at the adductor pollicis or the forehead, respectively) were measured and kept above 36 °C and 34 °C, respectively, by warm blankets or a convective warming device (Bair Hugger^®^ 505, Augustine Medical. Inc., MN, USA).

### 2.4. Groups

Patients were allocated to 2 groups:

Group A:

In the first 25 patients, measurements were performed simultaneously at the M. adductor pollicis of both sides. The allocation of the TOF 3D to the right or left arm was randomized by a computerized allocation schedule. In this group, the intravenous line was inserted into a vein of the forearm to exclude the interference of the intravenous line with the positioning of the hand or the muscle response. Both arms were carefully fixed on a splint.

Group CS:

In the second group, measurements were performed simultaneously at the M. corrugator supercilii of both sides. Again, the allocation of the TOF 3D to the right or left muscle was randomized by a computerized allocation schedule.

### 2.5. Neuromuscular Measurements

Following induction of anesthesia and laryngeal mask insertion, the neuromuscular transmission was assessed by simultaneous monitoring of acceleromyographic responses with the TOF 3D (Mammendorfer Institut für Physik und Medizin, Munich, Germany) and the TOF Watch SX (Essex Pharma GmbH, Munich, Germany) recorded from the right and the left M. adductor pollicis (Group A) or the M. corrugator supercilii (Group CS), respectively. Electrical stimulation of the nerves was achieved using transcutaneous Ag/AgCl electrodes (electrocardiogram electrodes; Ambu Inc., MD, USA) placed over the ulnar nerve at the wrist, or the facial nerve, respectively. The distance between the stimulating electrodes was 2 cm. Stimulation of the ulnar nerve was quantified by measurement of the thumb response. To minimise movement-induced changes in the response amplitude of the adductor pollicis, the hand was carefully fixed with tape to an arm board while the thumb was free to move [4]. Additionally, a commercially available preload device provided by the manufacturer (Hand Adapter^®^, Mammendorfer Institut für Physik und Medizin, Munich, Germany) was used to ensure an identical initial position of the thumb after each measurement. Due to the considerable distance between the ulnar nerve stimulating electrodes and the thumb, direct stimulation of adductor pollicis was considered to be unlikely. Appropriate stimulation of the facial nerve was verified by the response of the ipsilateral eyebrow. The positive electrode was placed proximally below the ear, while the negative electrode was positioned distally in front of the ear. The head was placed in a neutral position [4].

For acceleromyographic monitoring of muscle contractions, the accelerometer’s sensor was fixed to the distal portion of the thumb [4], or at the eyebrow. To establish a control twitch height value of 100%, the acceleromyographs were calibrated to a delivered supramaximal train-of-four (TOF) stimulus (2 Hz every 15 s, pulse width 200 µs). The acceleromyographs possess a calibration function that automatically determines the individual supramaximal stimulation current (up to a maximum current of 60 mA). The maximal acceleromygraphic response is automatically stored and serves as a reference control value for all subsequent measurements [9]. The first of the four twitch height responses was considered to be T_1_, and the TOF ratio was the ratio of the fourth twitch (T4) height response compared to T_1_. During the first minutes of stimulation, the acceleromyographic signal frequently drifts. Therefore, after a 10 min period of stabilisation, both acceleromyographs were recalibrated and 10 consecutive T_1_ and TOF ratio values simultaneously recorded at both measurement sites served as control values to assess the repeatability of the measurements. In group CS, simultaneous stimulation might have led to the interference of the devices because of their proximity to the forehead. Therefore, stimulation of the right side was started 5 s after stimulation of the left side in all cases, irrespective of the placement of the devices.

Following acceleromyographic calibration, rocuronium 0.6 mg kg^−1^ iv was injected (over 5 s) and the intravenous tubing was flushed with Ringers’ solution. Afterward, the following parameters were measured simultaneously at both locations: maximum T_1_ depression; onset time of neuromuscular block (i.e., the time from the beginning of rocuronium injection and maximum T_1_ depression); post-tetanic count (PTC), time to 25% T1 recovery (i.e., time from the start of rocuronium administration to recovery of a 25% T1 twitch height); time to 90% TOF recovery (i.e., time from the start of rocuronium injection to a TOF ratio of 0.9) [4]. Additionally, the time courses from the injection of rocuronium to the first reoccurrence of T_1_, T_2_, T_3_, and T_4_, respectively, and to full recovery of T_1_ and the TOF ratio were measured. All data were continuously recorded on a laptop connected to each device. In cases of varying values, the endpoint was regarded as the first of three consecutive T_1_ or TOF responses with the same or increasing amplitude [4].

### 2.6. Surgical Rating Scale

The surgeon assessed the quality of the surgical conditions by means of a standardized score: the Surgical Rating Score (SRS), introduced by Martini et al. and is commonly used for this purpose [7,8]. Its scale ranges from 1 (extremely poor conditions) to 5 (excellent conditions). Assessment of the quality of surgical conditions was performed by the surgeon without information with regard to the actual depth of the neuromuscular block or the drugs administered for improvement of the block. Measurements were performed in case of deterioration of the SRS. Additionally, the SRS was assessed at the following time points: measurement of PTC, recovery to T_2_, T_4_, and TOFR > 90%, respectively.

### 2.7. Statistical Analysis

Statistical analysis was performed using Sigma Plot 12.3 for Windows software package (Systat Software Inc., Chicago, IL, USA). In each patient, 10 consecutive measurements of the T_1_ and TOF ratio with both devices were simultaneously recorded prior to injection of rocuronium. The repeatability of these measurements was analyzed by means of a one-way analysis of variance (ANOVA). For all patients, the individual repeatability coefficients (calculated as 1.96 √2SD) were compared between both measurement sites with a paired t-test [4,5,6,10]. Additionally, the onset and time course of the neuromuscular block was compared between both devices by means of a paired t-test. These responses were used to calculate their differences and limits of agreement according to Bland and Altman [6].

Mean and standard deviation was calculated for patients’ characteristics and initial rocuronium dose.

Unless otherwise indicated, values are presented as means (standard deviation (SD)). The study sample size was calculated assuming a power of 80% to detect a 5% difference in time to 90% TOF recovery with a type 1 error of 0.05 and an anticipated TOF ratio SD of 8%; accordingly, 23 patients were required. In order to compensate for possible dropouts and because the expected SD was only estimated, 50 patients were enrolled (25 patients for each group) [11].

## 3. Results

### 3.1. Patient Characteristics

We enrolled all 50 patients in the analysis. They were 48.1 ± 11.4 years old, weighed 71.2 ± 14.2 kg, mean height was 167.7 ± 6.1 cm (mean ± SD), median SRS scores were 5 (5/5) (median (IQR)), and mean BIS scores ranged from 44.0 ± 6.1 to 48.6 ± 13.9 (mean ± SD). Differences between group A and CS or during the time points of measurements were not observed. A flow chart of patient distribution is provided in Figure 1.

### 3.2. Baseline Responses, Repeatability

The groups did not differ with regard to stimulation current and mean baseline TOFR. In group A, the repeatability coefficient of the TOF Watch SX measurements was similar with 4.08 ± 1.3 compared to 3.95 ± 1.2 measured by the TOF 3D (mean ± SD). However, in group CS, the repeatability coefficient measured by the TOF 3D was 4.66 ± 1.6 and therefore lower than that measured by the TOF Watch SX (6.02 ± 1.9; *p* = 0.026) (Table 1).

### 3.3. Onset and Maximal Neuromuscular Block, PTC

In group A, the onset of the block was faster when measured by the TOF 3D (98.7 ± 30 s vs. 112.2 ± 36 s (mean ± SD); *p* = 0.032). In group CS, both devices measured similar onset times (TOF Watch SX: 85.4 ± 31 s; TOF 3D: 86.8 ± 42 s). No differences between the devices were recorded with regard to maximal twitch depression or PTC.

### 3.4. Time Course of the Neuromuscular Block

In group A, the reappearance of T_1_–T_4_ and recovery to DUR 25% was faster when measured with the TOF 3D. In group CS, this could only be observed for the reappearance of T_2_–T_4_ (Table 2). Figure 2 shows the time course of the TOFR for both acceleromyographs in group A (Figure 2a) and group CS (Figure 2b). In group A, the recorded TOFR measured by the TOF 3D was higher 30 and 35 min after injection of rocuronium when compared to the TOF Watch SX.

### 3.5. Agreement

Onset: In group A, onset was earlier when measured with the TOF 3D compared to the TOF Watch SX (bias −0.23 min), with limits of agreement from −1.22 to +0.76 min. In group CS, onset was similar (bias + 0.023 min), with limits of agreement ranging from −1.52 to +1.56 min (Figure 3a).

Time to 25% T_1_ recovery (DUR 25%): In group A, recovery was faster when measured with the TOF 3D (bias: −2.14 min), with limits of agreement ranging from −7.32 to +3.04 min. In group CS, a bias of −1.88 min and limits of agreement ranging from −12.63 to +8.87 min were recorded with the TOF 3D (Figure 3b).

Time to recovery to TOFR 0.9: In group A, time to recovery to TOFR 90% was measured earlier by the TOF 3D (bias −0.71 min), with limits of agreement ranging from −8.94 to +7.51 min. In group CS, TOFR 90% was recorded earlier (bias: −3.85 min) with the TOF 3D. Limits of agreement ranged from −32.66 to +24.95 min in this group (Figure 3c).

## 4. Discussion

In the present study, the precision of the TOF 3D was high compared with the TOF Watch SX. The repeatability coefficient was lower than 5% at both measurement sites and therefore met the criteria postulated by other authors [12]. Additionally, it lay within the range of the results of previous studies [10,12,13,14]. The TOF 3D seemed to be more sensitive than the TOF Watch SX, since recovery was faster in nearly 60% of the measurements, which might be explained by the new three-dimensional sensor, enabling it to record movements in all directions.

At the M. adductor pollicis, both devices showed good agreement: when twitch height had recovered to 25% (DUR25%), a bias of 5% with limits of agreement of 12–15% were recorded. At the time to recovery to TOFR 90%, bias was ±1–2%, with limits of agreement of 14%. These observations correspond well with the results from other authors measuring the M. adductor pollicis: Dubois et al. reported a bias of 3% together with limits of agreement of 10–20%, and Claudius et al. observed a bias of 10% and limits of agreement of 20–30%, both comparing acceleromyography and mechanomyography [10,12]. Additionally, Motamed et al. found a bias of 1% and limits of agreement of 20% in a study assessing kinemyography and acceleromyography [15].

With regard to the onset of the block, limits of agreement were substantially higher reaching nearly 60%. However, this phenomenon has already been reported previously: while Claudius et al. reported limits of agreement of 40%, Dubois et al. and Motamed et al. observed 48% and 49% assessing the onset of the block [10,15].

At the M. corrugator supercilii, measurements had a faster onset and a shorter time to recovery compared with the M. adductor pollicis. These observations match well with the different onset and recovery profiles of both muscles [3]. Additionally, they might relate to the fact that the M. corrugator can be stimulated directly by the acceleromyograph. In these cases, muscle contractions are observed even in the presence of a deep or moderate neuromuscular block. Moreover, the movement of the eyebrow is shorter than that of the thumb, which might explain the lower precision of the measurements irrespective of the measuring device. However, measurement at the M. corrugator supercilii also offers advantages: The site is accessible in most cases when neuromuscular blocking agents are administered (e.g., during laparoscopic operations). Additionally, this muscle is a better indicator of laryngeal relaxation and it reflects the relaxation of abdominal muscles better than the M. adductor pollicis [16,17,18]. The distinct differences between these muscles were the reason why we decided to measure at the M. adductor pollicis and the M. corrugator supercilii, but did not compare the results of both measurement sites directly with each other.

At the M. corrugator supercilii, the repeatability coefficient of the TOF 3D was lower than that of the TOF Watch SX, and within the range of 5% mentioned above [12], while the TOF Watch SX did not achieve this target. During the time course of the neuromuscular block, bias and limits of agreement were substantially higher at the M. corrugator supercilii than at the M. adductor pollicis. However, many of the results of the measurements were similar at both measurement sites. Thus, if the M. corrugator supercilii is chosen for the assessment of a neuromuscular block, the new device TOF 3D might offer advantages because of its higher precision.

A strength of the study is the assessment at two different measurement sites: several differences were only observed at one of the measurement sites: the lower repeatability coefficient of the TOF 3D could only be observed at the M. corrugator supercilii. On the other hand, the differences with regard to the time course of the block were more pronounced at the M. adductor pollicis.

We were not able to detect differences with regard to the SRS score between the devices irrespective of the degree of the block or the measurement site. Two explanations might be given for these observations: (i) The degree of the neuromuscular block measured during our study did not differ widely between both devices. In this situation, a subjective evaluation such as the SRS score might not be able to discriminate differences better than the objective neuromuscular measurement. (ii) The SRS score assesses surgical conditions, not the degree of a neuromuscular block. However, surgical conditions might be good or excellent even without neuromuscular block, if analgesia and sedation are adequate.

Many investigators use normalized data for the assessment of the neuromuscular block. With this approach, the observed TOFR during recovery is corrected in relation to the baseline values. Bias and limits usually are wider with non-normalized data [13,14,19]. Additionally, TOFR baseline values often exceed 100% without normalization, which was the case in our investigation. However, measurements in the clinical setting are usually performed without normalization or even calibration prior to administration of the neuromuscular blocking agent, therefore we decided to use non-normalized data. Nevertheless, we were able to record acceptable data with regard to onset, time course, precision, bias, and limits of agreement. Moreover, calibrated and non-calibrated data show good agreement [20]. Therefore, our results probably are relevant for the anaesthesiologists in the clinical routine.

## 5. Conclusions

The new TOF 3D provides adequate information with regard to the degree of the neuromuscular block with high precision and sensitivity. It is suitable even for measurement sites with low-amplitude muscle contractions such as the M. corrugator supercilii. Therefore, it might serve as an appropriate alternative for the old ‘gold standard’ TOF Watch SX, which is not produced any longer.

## Figures and Tables

**Figure 1 jcm-11-04583-f001:**
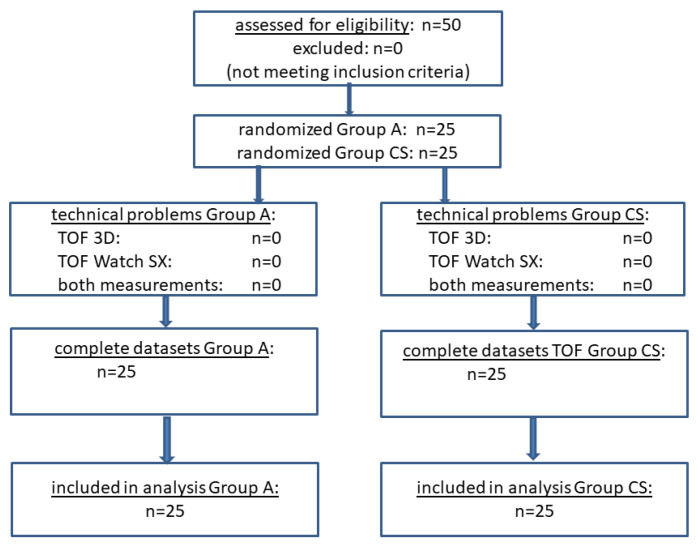
Consort flow diagram of the patients.

**Figure 2 jcm-11-04583-f002:**
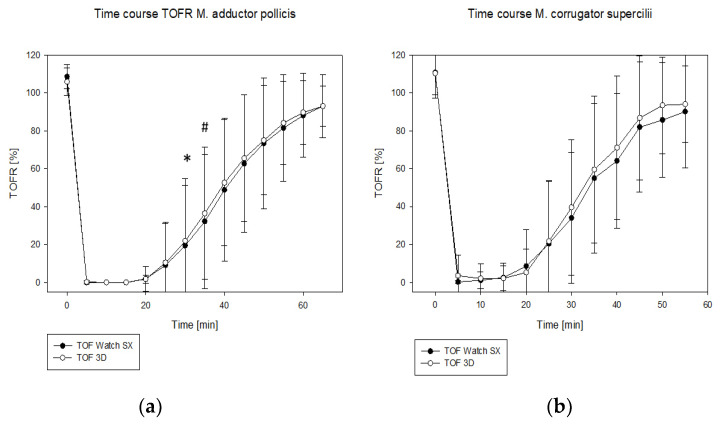
(**a**,**b**) Time course of train-of-four ratio (TOFR) vs. time for both acceleromyographs at the M. adductor pollicis (**a**) and the M. corrugator supercilii (**b**). Rocuronium was injected at time “0”. Black dots: TOF Watch SX. White dots: TOF 3D. *: *p* = 0.043 for 3D vs. SX. #: *p* = 0.0028 for 3D vs. SX.

**Figure 3 jcm-11-04583-f003:**
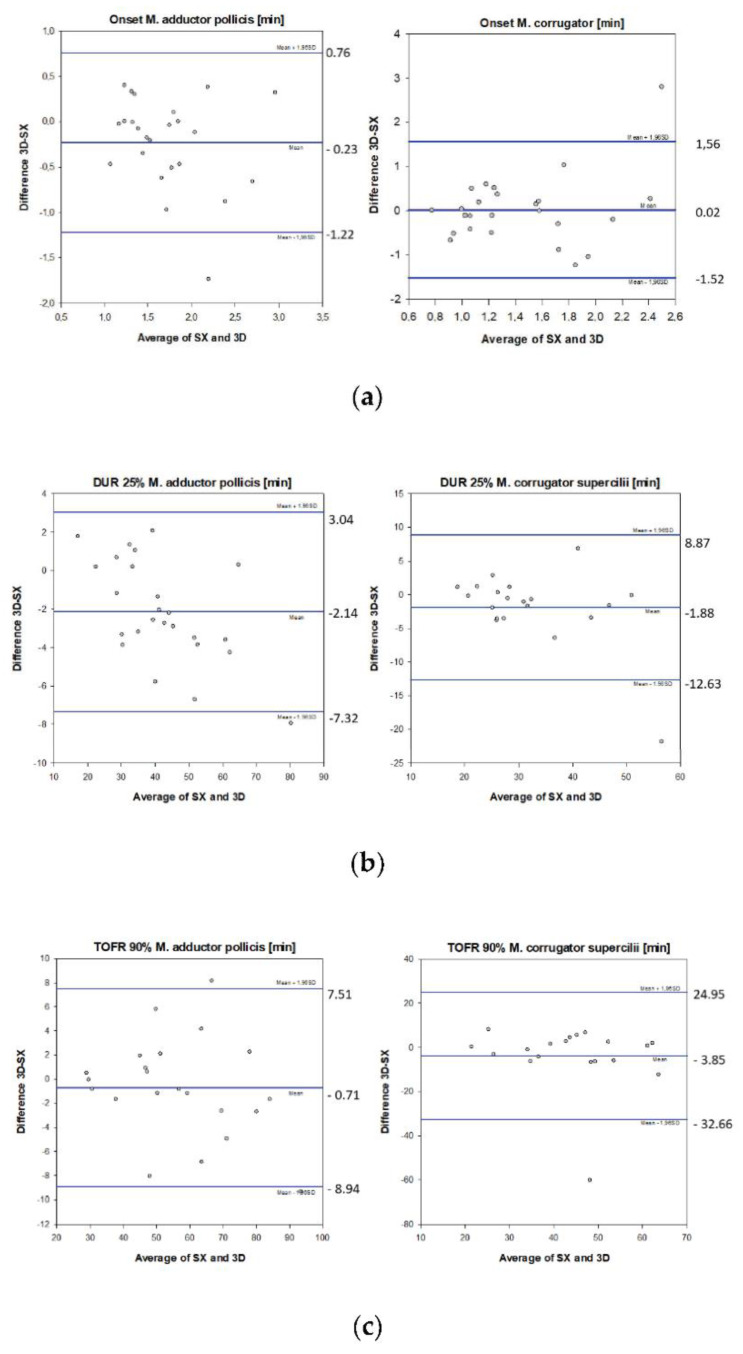
(**a**–**c**). Bland-Altman plots of the differences between TOF Watch SX and TOF 3D. Black horizontal line in the middle: bias; blue upper and lower line: limits of agreement (±1.96 SD); (**a**) Onset: time (min) between the beginning of injection of rocuronium and maximum T_1_ depression; (**b**) DUR25%: time (min) between administration of rocuronium and recovery to 25% twitch height; (**c**) TOFR 90%: time (min) from an injection of rocuronium to 90% train-of-four recovery.

**Table 1 jcm-11-04583-t001:** Baseline responses.

Group	n	Stimulation (mA) TOF Watch SX	Stimulation (mA) TOF 3D	Baseline TOFR (%) TOF Watch SX	Baseline TOFR (%) TOF 3D	Repeatability Coefficient TOF Watch SX	Repeatability Coefficient TOF 3D
M. adductor	25	48.8 ± 10.3	48.5 ± 10.0	108.8 ± 6.1	106.5 ± 5.7	4.08 ± 1.3	3.95 ± 1.2
M. corrugator	25	48.0 ± 14.5	41.5 ± 9.7	109.3 ± 13.5	113.0 ± 9.9	6.02 ± 1.9	4.66 ± 1.6 *

TOFR: Train-of-four Rate; *: *p* = 0.026 for SX vs. 3D. Values are presented as mean ± SD.

**Table 2 jcm-11-04583-t002:** Time course of the neuromuscular block: Reappearance of T_1_ –T_4_, DUR 25%, TOFR 90%, TOFR 100%. Values are presented in min ± SD.

Group	T_1_ SX	T_1_ 3D	T_2_ SX	T_2_ 3D	T_3_ SX	T_3_ 3D	T_4_ SX	T_4_ 3D	DUR 25% SX	DUR 25% 3D	TOFR 90% SX	TOFR 90% 3D	TOFR 100% SX	TOFR 100% 3D
M. adductor	26.3 ± 10	24.9 ± 8.9	31.5 ± 11.4	29,7 ± 10.2	34.7 ± 12.4	32.9 ± 10.9	36.1 ± 14.0	34.1 ± 11.2	42.1 ± 15.4	39.9 ± 14.0	59.1 ± 21.8	55.3 ± 17.4	61.0 ± 21.6	61.5 ± 19.9
P value for SX vs. 3D	0.005		0.005		0.003		0.011		< 0.001					
M. corrugator	17.0 ± 9.5	15.4 ± 8.9	23.3 ± 9.1	22.9 ± 7.7	27.3 ± 10.2	26.2 ± 8.4	36.1 ± 14.0	34.1 ± 11.2	31.5 ± 12.2	31.3 ± 9.4	45.0 ± 14.8	43.7 ± 14.7	45.7 ± 14.0	47.7 ± 13.8
P value for SX vs. 3D			0.043		0.039		0.011							

T_1_–T_4_: Time between injection of rocuronium and reappearance of T_1_, T_2_, T_3_ or T_4_; DUR25%: time between injection of rocuronium and recovery to 25% twitch height; TOFR 90% and TOFR 100%: time between injection of rocuronium to 90% or 100% train-of-four recovery, respectively.

## Data Availability

Not applicable.

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
