# Peer review of "Assessment of the New Acceleromyograph TOF 3D Compared with the Established TOF Watch SX: Bland-Altman Analysis of the Precision and Limits of Agreement between Both Devices—A Randomized Clinical Comparison"

_jcm, 2022, doi:10.3390/jcm11154583_

Round 1
Reviewer 1 Report
This is an interesting study comparing the performance of two train of four devices for the assessment of N euro-muscular block at two different sites in anaesthetised gynaecological patients.
A sentance re the origins and potential advatages of TOF measurement would add to the Introduction.
It appears that one single dose of Rocuronium was administered It may well have been a greater test/comparison of the devices if multiple doses had been administered and id ,as required the block had been reversed
LINES 70-73 It is freable to include full details of the patients here rather than add some in LINES 201-205.
LINE 89 " Normocapnia was achieved" --How was it maintained?
LINES 154-180-Surgical assment of relaxation- This section should be shortened as it contributes very little to the main aims of the paper .It could be covered by a suitable reference and short explanation.
LINE 255 "Gold standard"! what exactly is meant by this term?
Line 257 word "lied" -replace by lay.
Reviewer 2 Report
The authors present a well-designed study evaluating a new AMG device by comparing its performance with its predecessor at 2 muscle groups. The design is sound, the figures and tables are clear, and the text does an excellent job of conveying the results. I had heard that a new version of TOFWatch was coming and this is my first exposure to literature evaluating it. Quantitative neuromuscular monitoring is a trending topic and the authors should be commended for releasing this timely research project. I have only the following minor comments.
Methods: The authors describe fixing the hand to an arm board. Was a preload device utilized? If not, why? Was the goal to mirror real-world, clinical practice? This has been shown to significantly improve precision with AMG at the hand.
P8L255: State the main result in a complete sentence rather than opening the discussion with a the heading "main results:..."
P8L255: I would remove "gold standard". While the TOFwatch has been well described in the literature, I have never heard it referred to as the 'gold standard"
P8L259: Add numeric results to the statement "...faster in many measurement...". What percentage of measurements?
Discussion: The paragraph addressing the different muscle sensitivities between CS and AP fails to mention that these muscles have different NMB onset and recovery profiles, independent of the monitor utilized. Yes, direct stimulation is a risk when monitoring at the face, but even if electrodes are placed proximally on the facial nerve (just in front of the tragus and below the ear), CS will still become paralyzed sooner and recover faster than AP. Also, the results section needs to go into more specifics about where the stimulating electrodes were placed on the face. The authors should utilize one of the references they have previously used to describe placement (#4).
Discussion: I understand the comments regarding normalization vs non-normalized values. I agree that clinical practice does not routinely normalize these values. Nonetheless, the authors should expand these discussion to address the reverse fade phenomenon as this explains why many of their baseline values exceeded 100%. Preload application has been shown to mitigate this phenomenon, which can be another talking point.
Nonetheless, the authors are to be commended for a tremendous effort.
